# Sensitive Detection of Ciguatoxins Using a Neuroblastoma Cell-Based Assay with Voltage-Gated Potassium Channel Inhibitors

**DOI:** 10.3390/toxins16030118

**Published:** 2024-02-29

**Authors:** Toshiaki Yokozeki, Madoka Kawabata, Kazuhiro Fujita, Masahiro Hirama, Takeshi Tsumuraya

**Affiliations:** 1Osaka Saito Laboratory, Japan Food Research Laboratories, 7-4-41 Saitoasagi, Osaka 567-0085, Ibaraki, Japan; yokozekit@jfrl.or.jp (T.Y.); fujitak@jfrl.or.jp (K.F.); 2Department of Biological Chemistry, Graduate School of Science, Osaka Metropolitan University, 1-2 Gakuen-cho, Naka-ku, Sakai, Osaka 599-8570, Kanagawa, Japanmasahiro.hirama.c2@tohoku.ac.jp (M.H.)

**Keywords:** ciguatoxins, neuroblastoma cell-based assay, voltage-gated sodium channels, voltage-gated potassium channels, 4-aminopyridine, tetraethylammonium chloride

## Abstract

Ciguatoxins (CTXs) are neurotoxins responsible for ciguatera poisoning (CP), which affects more than 50,000 people worldwide annually. The development of analytical methods to prevent CP is a pressing global issue, and the N2a assay is one of the most promising methods for detecting CTXs. CTXs are highly toxic, and an action level of 0.01 μg CTX1B equivalent (eq)/kg in fish has been proposed. It is desirable to further increase the detection sensitivity of CTXs in the N2a assay to detect such low concentrations reliably. The opening of voltage-gated sodium channels (Na_V_ channels) and blocking of voltage-gated potassium channels (K_V_ channels) are thought to be involved in the toxicity of CTXs. Therefore, in this study, we developed an assay that could detect CTXs with higher sensitivity than conventional N2a assays, using K_V_ channel inhibitors as sensitizing reagents for N2a cells. The addition of the K_V_ channel inhibitors 4-aminopyridine and tetraethylammonium chloride to N2a cells, in addition to the traditional sensitizing reagents ouabain and veratridine, increased the sensitivity of N2a cells to CTXs by up to approximately 4-fold. This is also the first study to demonstrate the influence of K_V_ channels on the toxicity of CTXs in a cell-based assay.

## 1. Introduction

Ciguatoxins (CTXs) are neurotoxins with a ladder-like polyether structure produced by dinoflagellates of the genera Gambierdiscus [1,2,3] and Fukuyoa [4]. CTXs are transferred through the food chain from herbivorous to carnivorous fish [5], ultimately threatening the food supply. Globally, over 50,000 people suffer from these toxins annually, resulting in a global food-borne illness called ciguatera poisoning (CP) [6]. The main symptoms include gastrointestinal, cardiovascular, and neurological disorders, in severe cases, which can last for months or longer [7]. The main vectors of CTXs are coral reef fish in tropical and subtropical regions. However, in recent years, it has been reported that CTXs are also present in deep-sea fishes [8] and invertebrates [9,10,11], suggesting that these marine organisms are also part of the CTX food web. The habitat of dinoflagellates will change owing to global warming [12], and since the 2000s, the risk of CP has expanded to the North Atlantic [13,14]. In addition, CP occasionally occurs in fish imported into Europe [15,16,17,18]. Therefore, establishing preventive measures to identify food contaminated with toxins is of major interest.

CTXs are classified into Pacific, Caribbean, and Indian Ocean types, based on their basic structure. However, this is a nomenclature classification, and it has been reported that the classified CTXs are not restricted to their oceanic region [19,20]. CTXs from the Pacific undergo bio-oxidation in the food chain with increased toxicity [21,22], and the number of congeners reaches >20 [23]. For the Caribbean CTXs (C-CTXs), only the structures of C-CTX-1/-2 isolated from fish have been long clarified [24]; all the dinoflagellates producing C-CTXs have not been identified. However, recent structural analyses using liquid chromatography–high resolution mass spectrometry (LC-HRMS) have finally revealed the structures of new C-CTX congeners in fish [25,26] and a precursor of C-CTX-1/-2 in dinoflagellates [27]. On the other hand, the structures of the six compounds found in the Indian Ocean have not yet been elucidated [28,29].

The United States Food and Drug Administration (FDA) has proposed that CTX concentrations of 0.01 μg CTX1B equivalent (eq)/kg and 0.1 μg C-CTX-1 eq/kg in fish flesh are unlikely to cause CP [30]. As can be seen from these proposal levels, CTXs are highly toxic. In addition, ciguateric fish cannot be judged based on taste or appearance. Therefore, to prevent CP, it is essential to develop highly sensitive analytical methods to detect CTXs at such low levels in fish. The mouse bioassay (MBA), which has been used for a long time for CTXs’ detection, has become obsolete owing to its insufficient detection capability and ethical concerns [31]. Therefore, alternative methods, such as liquid chromatography with tandem mass spectrometry (LC-MS/MS) [32,33], capillary LC-HRMS [25], sandwich enzyme-linked immunosorbent assay (ELISA) [34], receptor binding assay (RBA) [32,35,36,37], neuroblastoma cell-based assay (N2a assay) [32,36,38,39], and even portable biosensors [40,41,42,43] have been developed in recent years. The RBA and N2a assays, which respond indistinguishably to all CTXs, complement LC-MS/MS and sandwich ELISA, which quantify the congeners individually. The European Food Safety Authority (EFSA) recommends that ciguateric fish should be screened using RBA or N2a assays and confirmed by LC-MS/MS [44], and the quantitative values of these methods have been shown to correlate with each other [45].

In 1984, one of the CTXs was shown to act on voltage-gated sodium channels (Na_V_ channels) to depolarize cells [46]. The N2a assay using this effect of CTXs was first reported in the 1990s [47,48,49]. Specifically, the addition of CTXs to mouse neuroblastoma cells (N2a cells) treated with ouabain (O) (Na^+^/K^+^-ATPase inhibitor) and veratridine (V) (Na_V_ channel opener) increases intracellular Na^+^ levels and induces CTX concentration-dependent cell death. Viable cells are then visualized using colorimetric reagents 3-(4,5-dimethylthiazol-2-yl)-2,5-diphenyltetrazolium bromide (MTT) or 2-(2-methoxy-4-nitrophenyl)-3-(4-nitrophenyl)-5-(2,4-disulfophenyl)-2H-tetrazolium, monosodium salt (WST-8), which are reduced by dehydrogenase in living cells [50]. We recently showed that the relative potency of CTX congeners against N2a cells correlates well with toxic equivalency factors (TEFs) based on acute toxicity in mice using quantitative nuclear magnetic resonance (NMR)-calibrated CTX congeners [36]. Since ciguateric fish typically contain multiple CTX congeners, the N2a assay can evaluate the overall toxicity of CTXs in fish in the same manner as MBA. Furthermore, in the same study, the total assay time for the N2a assay, which previously required >48 h, was reduced to approximately 24 h without compromising the detection capability. Although our work has made the N2a assay more valuable as a screening method for CTXs, the remaining challenge with the N2a assay is the interference of matrix components derived from fish flesh. The degree of the interference of the matrix in the N2a assay depends on the lipid content of the fish, with fatty fish requiring additional purification steps [51]. Recently, cyclodextrin polymers have been reported to effectively remove fish-derived matrices [52]. Another effective way to reduce matrix interference is to dilute the test solution, but to do this while maintaining the detection capability, the detection sensitivity of CTXs in the N2a assay needs to be increased.

In the 2000s, electrophysiological experiments showed that CTX1B also acts on voltage-gated potassium channels (K_V_ channels) at concentrations that act on Na_V_ channels [53,54]. K_V_ channels open after Na^+^ influx into the cells, and the cells are repolarized by K^+^ efflux. Thus, the inhibition of transient “A-type” potassium currents (*I*_K(A)_) and delayed-rectifier potassium currents (*I*_K(DR)_) by CTXs contributes to cell depolarization. Currently, both the opening of Na_V_ channels and blocking of K_V_ channels are thought to be involved in the toxicity of CTXs [55]. However, cell-based assays have not demonstrated whether inhibition of K_V_ channels synergistically increases CTXs’ toxicity.

Therefore, the objectives of this study were: (1) to develop a more sensitive N2a assay to CTXs than the conventional assay using only O and V, and (2) to confirm the effect of K_V_ channel inhibition on the toxicity of CTXs. For these purposes, K_V_ channel inhibitors were used in the N2a assay. The degree of inhibition of *I*_K(A)_ and *I*_K(DR)_ at CTX concentrations that depolarize Na_V_ channels is only partial. Therefore, the cytotoxicity of CTXs to N2a cells is expected to be enhanced using K_V_ channel inhibitors to further strengthen their inhibition. 

We report for the first time that combining O, V, and K_V_ channel inhibitors enables the detection of six Pacific CTX congeners (CTX1B, CTX3C, CTX4A, 52-epi-54-deoxyCTX1B, 54-deoxyCTX1B, and 51-hydroxyCTX3C) with higher sensitivity than the conventional N2a assay using only O and V.

## 2. Results

### 2.1. Optimization of O, V, and K_V_ Channel Inhibitors’ (4-AP and TEA-Cl) Concentrations

In this study, 4-aminopyridine (4-AP) and tetraethylammonium chloride (TEA-Cl) were used as inhibitors of *I*_K(A)_ and *I*_K(DR)_, respectively. These inhibitors are the reference compounds used to demonstrate the inhibition of *I*_K(A)_ and *I*_K(DR)_ by CTXs [54]. First, the cytotoxicity of 4-AP, TEA-Cl, and their mixture (4-AP/TEA-Cl) on N2a cells was evaluated to minimize cell death induced by K_V_ channel inhibitors. Furthermore, considering the possibility of using K_V_ channel inhibitors in combination with O and V in the N2a assay, the cytotoxicity of K_V_ channel inhibitors in the presence of 31.3/3.13 μM O/V (the optimized concentration for the N2a assay that we previously reported [36]) was also evaluated. 

As shown in Figure 1a,b, approximately 20% cell death was caused by 2.5 mM 4-AP and 12.5 mM TEA-Cl. Based on these potency ratios, 4-AP/TEA-Cl was evaluated using a 1:5 mixture, and 2.5/12.5 mM 4-AP/TEA-Cl caused approximately 30% cell death (Figure 1c). In addition, even when O/V was mixed with K_V_ channel inhibitors, there was almost no difference in toxicity compared to when O/V was not used (Figure 1a–c). Therefore, the optimal O/V concentration when used with K_V_ channel inhibitors was determined to be 31.3/3.13 μM. For K_V_ channel inhibitors, concentrations that caused little cell death compared to controls (4-AP, 0.5 mM; TEA-Cl, 2.5 mM; 4-AP/TEA-Cl, 0.5/2.5 mM) and concentrations that caused 20–30% cell death (4-AP, 2.5 mM; TEA-Cl, 12.5 mM; 4-AP/TEA-Cl, 2.5/12.5 mM) were used in subsequent experiments.

### 2.2. Cytotoxicity of CTX3C against N2a Cells in the Presence of K_V_ Channel Inhibitors 

For CTXs to induce N2a cell death, the addition of O and V is essential. However, it is not known whether K_V_ channel inhibitors alone could also cause CTX-induced cell death; therefore, we confirmed this using CTX3C. The concentrations of K_V_ channel inhibitors were set to cause 20–30% cell death to confirm the results under more severe conditions. As shown in Figure 2, within the CTX3C concentration range where a dose–response curve could be plotted at 31.3/3.13 μM O/V, it was not possible to plot dose–response curves for CTX3C under any K_V_ channel inhibitor condition.

### 2.3. Cytotoxicity of CTX Congeners against N2a Cells in the Coexistence of O, V, and K_V_ Channel Inhibitors

Since the K_V_ channel inhibitors alone did not increase the sensitivity of CTX3C to N2a cells in contrast to O/V, the effects of the K_V_ channel inhibitors were examined in the presence of 31.3/3.13 μM O/V. When comparing the dose–response curves for CTX3C at concentrations of K_V_ channel inhibitors that induced little cell death, all curves with K_V_ channel inhibitors almost overlapped with the O/V-only curve (Figure 3). 

In contrast, at concentrations that cause 20–30% cell death (4-AP, 2.5 mM; TEA-Cl, 12.5 mM; 4-AP/TEA-Cl, 2.5/12.5 mM), shifts of the sigmoidal curves to lower concentrations were observed for all six CTX congeners compared to the O/V-alone condition. The degree increased in the order of TEA-Cl, 4-AP, and 4-AP/TEA-Cl, except for CTX1B and 51-hydroxyCTX3C (which showed almost no difference between 4-AP and 4-AP/TEA-Cl) (Figure 4). The half-maximal effective concentrations (EC_50_s) are listed in Table 1.

## 3. Discussion

This study focused on K^+^ efflux, whereas the conventional N2 assays focus only on Na^+^ influx. Using K_V_ channel inhibitors to increase the intracellular concentration of K^+^ may increase the sensitivity of N2a cells to CTXs.

First, we examined the effects of K_V_ channel inhibitors (4-AP and TEA-Cl) on N2a cells and clarified the following three points: (1) the K_V_ channel inhibitors caused the death of N2a cells at concentrations in the mM range, with a potency ratio of TEA-Cl and 4-AP of 1:5, which is approximately the concentration ratio that inhibits K^+^ currents in dorsal root ganglion neurons in electrophysiological experiments [54]; (2) the cytotoxicity of K_V_ channel inhibitors to N2a cells was not enhanced by the coexistence of 31.3/3.13 μM O/V, the optimized concentration used in our previous work [36]; and (3) N2a cells were insensitive to CTX3C upon stimulation with K_V_ channel inhibitors alone. These results suggest that the combined use of O/V and K_V_ channel inhibitors is an appropriate condition to increase the sensitivity of CTXs in the N2a assay. Under these conditions, both Na^+^ and K^+^ were likely to be retained in the cells at higher concentrations. As expected, high concentrations of K_V_ channel inhibitors (4-AP, 2.5 mM; TEA-Cl, 12.5 mM; 4-AP/TEA-Cl, 2.5/12.5 mM) increased the sensitivity of N2a cells to CTXs. The degree of shift of the sigmoidal curves to lower concentrations was similar for most congeners, with some exceptions, increasing in the order of TEA-Cl (1.3–1.6 fold), 4-AP (2.2–3.9 fold), and 4-AP/TEA-Cl (2.4–4.1 fold). The magnitude of the shifts and the relative potency of the congeners before and after the changes are noteworthy. The detection sensitivity of CTX1B and CTX3C in N2a cells with lower sensitivity to O and V (OV-LS N2a cells in the literature) was 1.3-fold and 2.6-fold higher than that of the original N2a cells, respectively [39]. In this study, the mixed conditions of TEA-Cl and 4-AP using the original N2a cells showed more significant improvements in detection sensitivity than those using OV-LS N2a cells. In addition, this literature showed that CTX1B was more sensitive than CTX3C in the original N2a cells, reflecting the order of TEFs, whereas these sensitivities were reversed in OV-LS N2a cells. In contrast, sensitization with K_V_ channel inhibitors maintained a relative potency similar to that of the original condition with only O and V. We recently reported that the relative potency of CTX congeners in the N2a assay with only O and V correlated well with TEFs [36]. Therefore, we emphasize that the sensitive N2a assay using K_V_ channel inhibitors also correlates well with TEFs and can accurately assess the overall toxicity of CTXs. Table 2 summarizes the relative potencies of the CTX congeners under the four conditions investigated in this study, along with the TEFs in Reference [44].

Furthermore, since low concentrations of K_V_ channel inhibitors (4-AP, 0.5 mM; TEA-Cl, 2.5 mM; 4-AP/TEA-Cl, 0.5/2.5 mM) did not increase the sensitivity of N2a cells to CTXs, the degree of shift of the sigmoidal curves to lower concentrations seems to be dependent on the concentration of K_V_ channel inhibitors. Therefore, further optimization is expected to improve the sensitivity of this assay. Higher sensitivity allows for a higher dilution of the test solution, thus eliminating the need for pretest purification steps. It has been reported that CP can occur at concentrations just above the FDA action level of 0.01 μg CTX1B eq/kg in fish flesh [56,57], but it is difficult to accurately measure such low concentrations in fish with high lipid content, which is likely to cause matrix interference in the N2a assay [51]. Therefore, this method may allow for easier purification and highly sensitive measurement in many fish species, regardless of their lipid content. However, unlike the assay using OV-LS N2a cells, which has already been confirmed to detect CTXs even in the presence of fish-derived matrices, this assay has not yet been validated using fish samples. The possibility that using K_V_ channel inhibitors may also increase sensitivity to matrices should be confirmed.

Another highlight of this study is that the cell-based assay demonstrated that inhibition of K_V_ channels increased the cytotoxicity of CTXs. This study could not determine whether the blockade of K_V_ channels at CTX concentrations that cause CP enhances the cytotoxicity of CTXs in concert with the opening of Na_V_ channels. However, the finding that the more significant inhibition of *I*_K(A)_ and *I*_K(DR)_, partially blocked by CTXs, enhanced the cytotoxicity of CTXs in N2a cells suggests the involvement of K_V_ channels in the toxicity of CTXs. In N2a cells, outward potassium currents are increased by O, which inhibits Na^+^/K^+^-ATPase; however, this effect can be canceled by 4-AP [58]. This seems to be one of the reasons why the combination of K_V_ channel inhibitors with O/V is effective in increasing the cytotoxicity of CTXs to N2a cells. Regarding this discussion on cytotoxicity, it should be noted that N2a cells were used in this study. This is because it has been pointed out that N2a cells are transformed cell lines and are not suitable for pharmacological studies. Their receptor expression levels differed significantly from those of the parental cell type [59]. For example, the expression level of Na_V_ channels in N2a cells is only 1/20 that in primary cultures of cerebellar granule neurons (CGNs), and N-methyl-D-aspartate (NMDA) receptors expressed in CGNs are absent in N2a cells. Thus, NMDA receptor-mediated extracellular Ca^2+^ influx resulting from the activation of Na_V_ channels by C-CTX-1 occurs in CGNs but not in N2a cells. In addition, brevetoxins, neurotoxins that share the binding site on Na_V_ channels with CTXs, have been reported to cause cell death mainly through this mechanism [60], indicating that Ca^2+^ influx through NMDA receptors plays an essential role in neuronal toxicity. Ca^2+^ influx via this mechanism also occurs upon the inhibition of K_V_ channels in CGNs by 4-AP [61]. Taken together, the use of K_V_ channel inhibitors should result in extracellular Ca^2+^ influx through NMDA receptors and contribute to cell death, but in this study using N2a cells lacking NMDA receptors, this mechanism is unlikely to contribute to the increased cytotoxicity of CTXs. Conversely, enhancing CTXs’ cytotoxicity by adding K_V_ channel inhibitors to primary cells expressing NMDA receptors may be more significant.

Recently, it has been reported that the combination of CTX3C and deltamethrin, an insecticide that prolongs the open state of Na_V_ channels, has synergistic effects on reducing the maximum peak inward sodium currents and hyperpolarizing the activation voltage of Na_V_ channels [62]. In this study, the effect of K_V_ channels on the toxicity of CTXs was observed, thus raising interest in the synergistic effect of CTXs and gambierol on CP. Gambierol is a ladder-shaped polyether produced by *Gambierdiscus toxicus* together with CTX3C and CTX4A [2,3,63] and is a potent inhibitor of *I*_K(A)_ and *I*_K(DR)_ [61,64], resulting in NMDA receptor-mediated Ca^2+^ influx [65]. Gambierol itself is not as toxic as CTXs [66], and its detection in fish flesh has not been reported; therefore, its involvement in CP is inconclusive. However, since gambieric acid, which is produced by *Gambierdiscus toxicus* as well as gambierol [67], has been detected in fish flesh together with CTXs [29,68], it is natural to consider the possibility that other polyether compounds may coexist with CTXs in fish. The results of this study suggest that the coexistence of gambierol and CTXs may increase the toxicity of CTXs owing to the ability of gambierol to block K_V_ channels. Therefore, verifying this possibility using cell-based assays and clarifying whether gambierol coexists with CTXs in fish will be necessary in the future.

## 4. Conclusions

This is the first study to apply K_V_ channel inhibitors in N2a assays. CTXs were detected with higher sensitivity than the conventional N2a assay using only O and V simply by adding K_V_ channel inhibitors to the assay. The effect increased in the order of TEA-Cl, 4-AP, and 4-AP/TEA-Cl, reaching a maximum of approximately four-fold. This new sensitive N2a assay allows fish flesh extracts to be more diluted before the assay, potentially simplifying the complex purification steps to remove fish-derived matrices. Furthermore, the relative potency of CTX congeners to N2a cells in this assay correlated well with TEFs based on acute toxicity in mice, as well as in the conventional O/V-only assay, making it suitable for accurate assessment of the overall toxicity of CTXs. This is also the first study to demonstrate the influence of K_V_ channels on the toxicity of CTXs in a cell-based assay, which brings further interest to elucidating the role of K_V_ channels in CP.

We plan to further optimize this new N2a assay and demonstrate that it can reliably detect fish flesh at the FDA action level of 0.01 μg CTX1B eq/kg.

## 5. Materials and Methods

### 5.1. Chemicals

Water was Milli-Q ultrapure grade with 18.2 MΩcm resistivity. 4-AP and TEA-Cl were purchased from FUJIFILM Wako Pure Chemical Industry, Ltd. (Osaka, Japan). Fetal bovine serum (FBS) was purchased from Thermo Fisher Scientific Inc. (Waltham, MA, USA). All other reagents were purchased from Nacalai Tesque Inc. (Kyoto, Japan).

### 5.2. Reference CTXs

CTX1B, CTX3C, CTX4A, 52-epi-54-deoxyCTX1B, and 51-hydroxyCTX3C were calibrated using quantitative NMR at Japan Food Research Laboratories [69]. Hirama and coworkers synthesized 54-deoxyCTX1B at Tohoku University [70] and it was calibrated using sandwich ELISA [34] with reference to quantitative NMR-calibrated CTX1B. 

### 5.3. N2a Assay Using K_V_ Channel Inhibitors

#### 5.3.1. Passaging of the Cell Line

The N2a cell line was obtained from the European Collection of Cell Cultures (EC89121404, ECACC, Salisbury, UK). N2a cells were maintained and passaged according to previously described procedures [36]. Briefly, N2a cells were cultured in RPMI 1640 medium containing 10% FBS, 1 mM L-glutamine, 1 mM sodium pyruvate, 100 units/mL penicillin, and 100 μg/mL streptomycin (N2a medium) at 37 °C in a 5% CO_2_ humidified atmosphere. The seeding density was 1 × 10^5^ cells mL^−1^ (for the assays after 2 days) or 5 × 10^4^ cells mL^−1^ (for the assays after 3 days) in 75 cm^2^ tissue culture flasks (Sumitomo Bakelite, Tokyo, Japan). The trypan blue-stained cells were subjected to a TC20^TM^ automated cell counter (Bio-Rad, Hercules, CA, USA) immediately before the assay to determine the number of viable cells and cell viability. When the cell viability was ≥85%, the assays were performed.

#### 5.3.2. Optimization of O, V, and K_V_ Channel Inhibitors’ (4-AP and TEA-Cl) Concentrations

4-AP and TEA-Cl were dissolved separately in the N2a medium, and two-fold serial dilutions from 20 mM to 0.1563 mM for 4-AP and from 200 mM to 1.563 mM for TEA-Cl were prepared in the same medium. In addition, 4-AP and TEA-Cl were dissolved together in the N2a medium at a ratio of 1:5, and two-fold serial dilutions from 4-AP/TEA-Cl = 20/100 mM to 0.156/0.7813 mM were prepared in the same medium. 

In addition to the K_V_ channel inhibitor-only solutions, mixtures of O/V (final concentration of 31.3/3.13 μM, previously optimized in [36]) and K_V_ channel inhibitors were also prepared. Briefly, stock solutions of O (10 mM in water) and V (1 mM in 10 mM HCl) were diluted with N2a medium to prepare a 62.5/6.25 μM O/V solution. Next, 4-AP and TEA-Cl were dissolved separately in the O/V solution, and then the solutions were diluted with the O/V solution to prepare 4-AP concentrations of 1.25 mM and 5 mM, and TEA-Cl concentrations of 6.25 mM and 25 mM. In addition, 4-AP and TEA-Cl were dissolved together in the O/V solution at a ratio of 1:5, and the solution was diluted with the O/V solution to produce 4-AP/TEA-Cl concentrations of 1.25/6.25 mM and 5/25 mM. 

The assays for these test solutions followed our previously described procedure [36]. Briefly, 100 μL of cell suspension in the N2a medium was seeded into 96-well cell culture plates (Corning, Corning, NY, USA) at 5 × 10^4^ cells/well. Next, 100 µL of the test solution was added to the corresponding wells, and the plates were incubated for 22 h at 37 °C and 5% CO_2_. Finally, 10 μL of Cell Counting Kit-8 (Dojindo, Kumamoto, Japan) was added to each well and the plates were incubated for 3–4 h at 37 °C and 5% CO_2_. Absorbance at 490 nm was measured using an iMark^TM^ microplate reader (Bio-Rad). The cell viability of the control wells (N2a cells only, no inhibitors added) was defined as 100%, and the cell viability for each inhibitor concentration was calculated. Statistical analyses were by one-way ANOVA with Dunnett’s multiple comparison tests. All data were considered statistically significant at *p* < 0.05.

#### 5.3.3. Cytotoxicity of CTX Congeners against N2a Cells in the Presence of K_V_ Channel Inhibitors and the Coexistence of O, V, and K_V_ Channel Inhibitors

4-AP and/or TEA-Cl were dissolved in the N2a medium to prepare 5 mM 4-AP, 25 mM TEA-Cl, and 5/25 mM 4-AP/TEA-Cl solutions. A stock solution of CTX3C in DMSO was then serially diluted (1:2) with these K_V_ channel inhibitor solutions to obtain CTX3C test solutions ranging from 0.994 to 63.6 pg/mL. 

In addition to K_V_ channel inhibitor-only test solutions, mixtures of 62.5/6.25 μM O/V and K_V_ channel inhibitor (O/V + K_V_) test solutions were also prepared. The 62.5/6.25 μM O/V solution was prepared as described in Section 5.3.2. Subsequently, 4-AP and/or TEA-Cl were dissolved in the O/V solution to prepare 5 mM 4-AP, 25 mM TEA-Cl, and 5/25 mM 4-AP/TEA solutions with 62.5/6.25 μM O/V. In addition, 1 mM 4-AP, 5 mM TEA-Cl, and 1/5 mM 4-AP/TEA-Cl solutions with 62.5/6.25 μM O/V were similarly prepared as lower concentration conditions of K_V_ channel inhibitors. At lower concentrations, only CTX3C was tested, whereas at higher concentrations, all the CTX congeners were tested. Each DMSO stock solution of CTXs (CTX1B, 6.5 ng/mL; CTX3C, 6.36 ng/mL; CTX4A, 27.55 ng/mL; 52-epi-54-deoxyCTX1B, 5.84 ng/mL; 54-deoxyCTX1B, 4.89 ng/mL; 51-hydroxyCTX3C, 4.53 ng/mL) was diluted 100-fold with 62.5/6.25 μM O/V alone (for comparison with the conventional condition) or O/V + K_V_ solutions, which were then serially diluted (1:2.5) nine times with the exact solutions. 

The test solutions for each condition were assayed in the same manner as in Section 5.3.2, to obtain dose–response curves for the CTX congeners under each inhibitor condition. The cell viability of the control wells (N2a cells + inhibitors, no CTXs added) was defined as 100%, and the cell viability for each CTX concentration was calculated. The EC_50_s were determined by curve fitting with a variable slope–four parameter logistic regression model using GraphPad Prism version 10.0.2 (GraphPad Software, San Diego, CA, USA), according to the following Equation (1): Y = Bottom + (Top − Bottom)/(1 + 10^((LogEC_50_ − LogX) × HillSlope))(1)

Statistical analyses for EC_50_s were by one-way ANOVA with Dunnett’s multiple comparison tests. All data were considered statistically significant at *p* < 0.05.

## Figures and Tables

**Figure 1 toxins-16-00118-f001:**
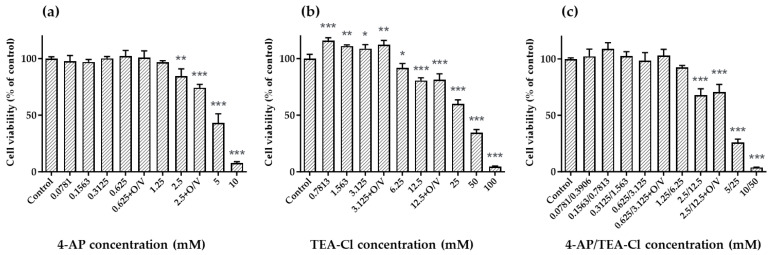
Cytotoxicity of K_V_ channel inhibitors against N2a cells: cytotoxicity of (**a**) 4-AP, (**b**) TEA-Cl, (**c**) mixture of 4-AP and TEA-Cl. +O/V indicates that 31.3 μM O and 3.13 μM V were added to N2a cells with K_V_ channel inhibitors. Data are the mean ± standard deviation (SD) of triplicate wells. * *p* < 0.05; ** *p* < 0.01; *** *p* < 0.001, significantly different from control.

**Figure 2 toxins-16-00118-f002:**
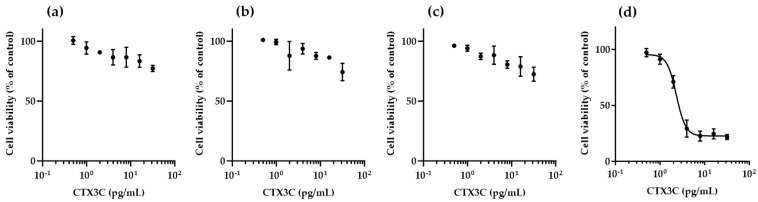
Cytotoxicity of CTX3C against N2a cells in the presence of K_V_ channel inhibitors: in the presence of (**a**) 4-AP, (**b**) TEA-Cl, (**c**) mixture of 4-AP and TEA-Cl, (**d**) 31.3/3.13 μM O/V (no K_V_ channel inhibitors added). Data are the mean ± SD of triplicate wells.

**Figure 3 toxins-16-00118-f003:**
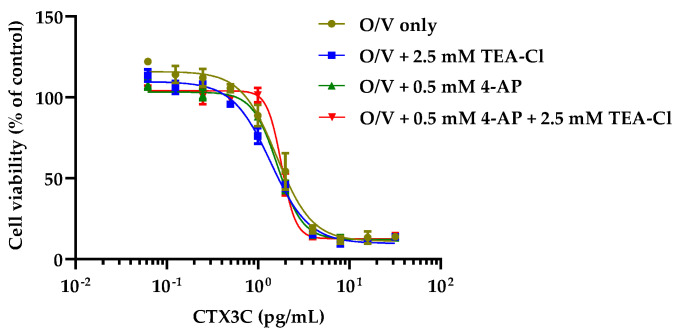
Cytotoxicity curves of CTX3C in the coexistence of O, V, and K_V_ channel inhibitors (low concentration conditions). Data are the mean ± SD of triplicate wells.

**Figure 4 toxins-16-00118-f004:**
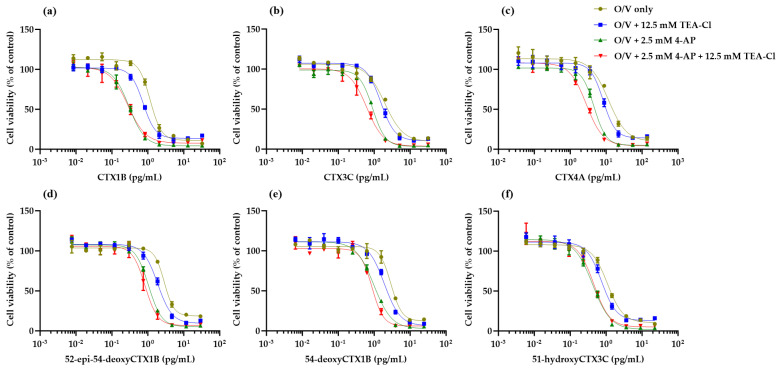
Cytotoxicity curves of CTX congeners in O/V alone and in the coexistence of O/V and K_V_ channel inhibitors (high concentration conditions): (**a**) CTX1B, (**b**) CTX3C, (**c**) CTX4A, (**d**) 52-epi-54-deoxyCTX1B, (**e**) 54-deoxyCTX1B, (**f**) 51-hydroxyCTX3C. Data are the mean ± SD of triplicate wells.

**Table 1 toxins-16-00118-t001:** EC_50_ values of CTX congeners calculated from dose–response curves.

Congeners	O/V Only (pg/mL)	O/V + TEA-Cl (pg/mL)	O/V + 4-AP (pg/mL)	O/V + 4-AP + TEA-Cl (pg/mL)
CTX1B	1.16 ± 0.09	0.74 ± 0.05 ***	0.30 ± 0.03 ***	0.28 ± 0.04 ***
CTX3C	1.96 ± 0.28	1.52 ± 0.11 *	0.90 ± 0.07 ***	0.62 ± 0.07 ***
CTX4A	11.7 ± 1.88	8.27 ± 0.78 **	4.59 ± 0.27 ***	2.82 ± 0.26 ***
52-epi-54-deoxyCTX1B	2.86 ± 0.23	1.88 ± 0.15 ***	1.05 ± 0.06 ***	0.84 ± 0.10 ***
54-deoxyCTX1B	2.91 ± 0.29	1.91 ± 0.20 ***	0.98 ± 0.07 ***	0.87 ± 0.09 ***
51-hydroxyCTX3C	1.09 ± 0.07	0.75 ± 0.08 ***	0.42 ± 0.04 ***	0.45 ± 0.06 ***

The values are the mean ± SD of triplicate wells. * *p* < 0.05; ** *p* < 0.01; *** *p* < 0.001, significantly different from O/V-only condition for each congener.

**Table 2 toxins-16-00118-t002:** Relative potency of CTX congeners in the four conditions.

Congeners	O/V Only	O/V + TEA-Cl	O/V + 4-AP	O/V + 4-AP + TEA-Cl	TEFs [44]
CTX1B	1.0	1.0	1.0	1.0	1.0
CTX3C	0.6	0.5	0.3	0.5	0.2
CTX4A	0.10	0.09	0.07	0.10	0.1
52-epi-54-deoxyCTX1B	0.4	0.4	0.3	0.3	0.3
54-deoxyCTX1B	0.4	0.4	0.3	0.3	0.3
51-hydroxyCTX3C	1.1	1.0	0.7	0.6	1.0

Each relative potency was calculated with CTX1B as 1.0.

## Data Availability

The data presented in this study are available on request from the corresponding author.

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
