# Peer review of "Sensitive Detection of Ciguatoxins Using a Neuroblastoma Cell-Based Assay with Voltage-Gated Potassium Channel Inhibitors"

_toxins, 2024, doi:10.3390/toxins16030118_

Round 1

Reviewer 1 Report

Comments and Suggestions for Authors

I have to congratulate the authors for their manuscript “Sensitive detection of ciguatoxins using a neuroblastoma cell- 2 based assay with voltage-gated potassium channel inhibitors” for their work. The paper at hand shows a modification of the classical N2a cell-based assay for CTX detection which improves its sensitivity allowing for a more accurate detection of CTX analogues and potentially reducing the time needed for sample preparation.  This is crucial for guaranteeing food-safety, particularly when dealing with fish products that are usually consumed fresh. The work also includes the evaluation of TEFs for each analogue with this improved method, which is also quite interesting. Furthermore, the manuscript is very well written and structured. Each section is clearly exposed, the tables and figures displayed are suitable to show the results obtained and the bibliographic references correctly chosen to support their arguments, with most of them being up-to-date and relevant for the field. As for the writing, I hardly found a typo in the text. Therefore, after an exhaustive revision, I think the paper can be published in present form. 

Author Response

Response to Reviewer 1 (reviewer’s comments are shown in blue):

I have to congratulate the authors for their manuscript “Sensitive detection of ciguatoxins using a neuroblastoma cell-2 based assay with voltage-gated potassium channel inhibitors” for their work. The paper at hand shows a modification of the classical N2a cell-based assay for CTX detection which improves its sensitivity allowing for a more accurate detection of CTX analogues and potentially reducing the time needed for sample preparation. This is crucial for guaranteeing food-safety, particularly when dealing with fish products that are usually consumed fresh. The work also includes the evaluation of TEFs for each analogue with this improved method, which is also quite interesting. Furthermore, the manuscript is very well written and structured. Each section is clearly exposed, the tables and figures displayed are suitable to show the results obtained and the bibliographic references correctly chosen to support their arguments, with most of them being up-to-date and relevant for the field. As for the writing, I hardly found a typo in the text. Therefore, after an exhaustive revision, I think the paper can be published in present form.

Thank you very much for your positive feedback concerning our manuscript. Since there were no comments from reviewer 1 regarding revisions, we carefully reviewed the manuscript again.

In addition to the revisions pointed out by the reviewers, we have tried to reduce the duplication rate as much as possible. They are highlighted in green. On the other hand, many of the duplicates are proper nouns related to experiments and used in previous paper published in Toxicon (2023). Please note that since this paper is an improvement of the method published in Toxicon, a certain amount of duplication is inevitable to ensure that the experiments are accurate.

We hope the revised manuscript is satisfactory for publication in Toxins. Thank you for your kind consideration.

Reviewer 2 Report

Comments and Suggestions for Authors

The Manuscript ID – toxins-2856554

Sensitive detection of ciguatoxins using a neuroblastoma cell-based assay with voltage-gated potassium channel inhibitors

Congeners, well applied term?

This study provides a comprehensive description of the use of Kv channel inhibitors 4-aminopyridine and tetraethylammonium chloride to N2a cells, in addition to the traditional sensitizing reagents such as ouabain and veratridine, which demonstrated to increase the sensitivity of N2a cells to CTXs. This contribution is important in the field of toxicology because demonstrate the influence of Kv channels on the toxicity of CTXs in a cell-based assay.

I consider that this study will be useful in the toxicology fields and good candidate for publication, since data are new and of interest.

Recommendations:

Paragraph 103-111 must be considered to move to the results and discussion sections.

Author Response

Response to Reviewer 2 (reviewer’s comments are shown in blue):

This study provides a comprehensive description of the use of KV channel inhibitors 4-aminopyridine and tetraethylammonium chloride to N2a cells, in addition to the traditional sensitizing reagents such as ouabain and veratridine, which demonstrated to increase the sensitivity of N2a cells to CTXs. This contribution is important in the field of toxicology because demonstrate the influence of Kv channels on the toxicity of CTXs in a cell-based assay.

I consider that this study will be useful in the toxicology fields and good candidate for publication, since data are new and of interest.

Thank you very much for your positive feedback concerning our manuscript.

Congeners, well applied term?

When we entered the words “Ciguatoxin congener” or “ciguatoxin homolog” on Google Scholar and searched for materials after 2020, we confirmed that “ciguatoxin congener” had a higher number of searches. Additionally, since we confirmed multiple papers in which it was used, we considered "congener" to be a well-applied term and did not make any changes to the paper.

Recommendations:

Paragraph 103-111 must be considered to move to the results and discussion sections.

Lines 103–105 were moved to Results 2.1. Lines 105–109 were removed because they are repeated in the Results. Lines 109–111 were left in the Introduction with some modifications (all CTXs → six Pacific congeners (CTX1B, …, and 51-hydroxyCTX3C)) because Toxins' Instructions for Authors states that the main conclusions should be highlighted in the Introduction.

In addition to the revisions pointed out by the reviewers, we have tried to reduce the duplication rate as much as possible. They are highlighted in green. On the other hand, many of the duplicates are proper nouns related to experiments and used in previous paper published in Toxicon (2023). Please note that since this paper is an improvement of the method published in Toxicon, a certain amount of duplication is inevitable to ensure that the experiments are accurate.

We hope the revised manuscript is satisfactory for publication in Toxins. Thank you for your kind consideration.

Reviewer 3 Report

Comments and Suggestions for Authors

Current Manuscript entitles “Sensitive detection of ciguatoxins using a neuroblastoma cell-based assay with voltage-gated potassium channel inhibitors” presents an innovative approach for the sensitive detection of ciguatoxins, employing a neuroblastoma cell-based assay with voltage-gated potassium channel inhibitors. This methodology holds great promise in enhancing the precision and sensitivity of ciguatoxin detection, offering valuable insights and potential advancements in the field of toxin assessment. The integration of neuroblastoma cells and potassium channel inhibitors demonstrates a novel strategy that could significantly contribute to the improvement of toxin detection methods. The manuscript can be recommended for publication after major changes and amendments.

With regard to the specific comment, I think this work provide novel approach for marine toxin detection, this is the main question the authors addressed. This method is promising and hold commerical prospects. But it still needs pretty long way to make it practical applicable due to the difficulties in maintain the activity of cells satble for a long time, especials for those in-field applications. But this is the main problem encounteded by all the cell-based biosensors, which could not be addressed by individual work. Anyway, I think this work could potentially contribute to this field.

1.      The introduction seems less, the author can discuss various analytical methods such as https://doi.org/10.1016/j.toxicon.2005.04.006,  https://doi.org/10.1016/j.scitotenv.2021.150915,   https://doi.org/10.1016/j.ecoenv.2020.111004

2.      The conclusion is written in a few lines, the authors should explain the conclusion more in detail.

3.      Acronyms and abbreviations need to be defined when they are first used.

4.      The authors need to recheck the whole manuscript to remove grammatical errors/ typos/ incomplete sentences and non-relative phrases.

Author Response

Response to Reviewer 3 (reviewer’s comments are shown in blue):

Current Manuscript entitles “Sensitive detection of ciguatoxins using a neuroblastoma cell-based assay with voltage-gated potassium channel inhibitors” presents an innovative approach for the sensitive detection of ciguatoxins, employing a neuroblastoma cell-based assay with voltage-gated potassium channel inhibitors. This methodology holds great promise in enhancing the precision and sensitivity of ciguatoxin detection, offering valuable insights and potential advancements in the field of toxin assessment. The integration of neuroblastoma cells and potassium channel inhibitors demonstrates a novel strategy that could significantly contribute to the improvement of toxin detection methods. The manuscript can be recommended for publication after major changes and amendments.

With regard to the specific comment, I think this work provide novel approach for marine toxin detection, this is the main question the authors addressed. This method is promising and hold commerical prospects. But it still needs pretty long way to make it practical applicable due to the difficulties in maintain the activity of cells satble for a long time, especials for those in-field applications. But this is the main problem encounteded by all the cell-based biosensors, which could not be addressed by individual work. Anyway, I think this work could potentially contribute to this field.

Thank you very much for your positive feedback concerning our manuscript.

1. The introduction seems less, the author can discuss various analytical methods such as https://doi.org/10.1016/j.toxicon.2005.04.006,

https://doi.org/10.1016/j.scitotenv.2021.150915,

https://doi.org/10.1016/j.ecoenv.2020.111004

As suggested, we added these three references to the text explaining the analysis methods for ciguatoxins (Reference No. 32, 42 and 43).

2. The conclusion is written in a few lines, the authors should explain the conclusion more in detail.

As suggested, we rewrote the conclusion more specifically and emphasized the important points of this paper (Lines 273–285).

3. Acronyms and abbreviations need to be defined when they are first used.

We added definitions to the following five acronyms and abbreviations.

Line 8, 51:equivalent (eq)

Line 72  :3-(4,5-dimethylthiazol-2-yl)-2,5-diphenyltetrazolium bromide (MTT)

Line 73  :2-(2-methoxy-4-nitrophenyl)-3-(4-nitrophenyl)-5-(2,4-disulfophenyl)-2H-

tetrazolium, monosodium salt (WST-8)

Line 77 :nuclear magnetic resonance (NMR)

Line 136   :standard deviation (SD)

4. The authors need to recheck the whole manuscript to remove grammatical errors/ typos/ incomplete sentences and non-relative phrases.

We thoroughly reviewed the manuscript and made the following changes, in addition to those suggested by other reviewers.

1) Nav and Kv in Abstract were defined (Lines 10–11).

2) The "+" in "Na+" was corrected to a superscript (Line 177).

In addition to the revisions pointed out by the reviewers, we have tried to reduce the duplication rate as much as possible. They are highlighted in green. On the other hand, many of the duplicates are proper nouns related to experiments and used in previous paper published in Toxicon (2023). Please note that since this paper is an improvement of the method published in Toxicon, a certain amount of duplication is inevitable to ensure that the experiments are accurate.

We hope the revised manuscript is satisfactory for publication in Toxins. Thank you for your kind consideration.

Reviewer 4 Report

Comments and Suggestions for Authors

The article detection of ciguatoxins using a neuroblastoma cell-based assay with voltage-gated potassium channel inhibitors is an exciting approach to a better understanding of the mode of action of ciguatoxins and utilizing this information for better sensitivity in samples for human safety. This paper must be accepted, but some minor changes are proposed:

NaV and KV channels are miswritten all along the document. They should have a capital V in subscript. Please change them.

CFP is no longer called CFP since the toxins have been identified in many other organisms, not only fish. Please check the recent references on the subject. Now, it is called only CP -- ciguatera poisoning.

L37- I suggest changing "problem" to "interest". Establishing preventive measures to identify food contaminated with toxins is of major interest. The measures are not "against" ciguatera, but against human intoxication. Please, reword the sentence.

L49-51 - this sentence is miswritten and not clear. Please, rewrite for clarification.

L116 - it is not "cell death induced by themselves"; it is auto-induced death.

L118 - I am not sure of the usefulness of two and three-decimal numbers. It might be easier to read 31.3/3.1

Figure 1. It is not clear what the x axis is referring to. Please clarify.

Materials and methods: it is preferable to use scientific notation (cells mL-1) than cells/mL

Comments on the Quality of English Language

There are few details on the English quality, but in general, it is good.

Author Response

Response to Reviewer 4 (reviewer’s comments are shown in blue):

The article detection of ciguatoxins using a neuroblastoma cell-based assay with voltage-gated potassium channel inhibitors is an exciting approach to a better understanding of the mode of action of ciguatoxins and utilizing this information for better sensitivity in samples for human safety. This paper must be accepted, but some minor changes are proposed:

Thank you very much for your positive feedback concerning our manuscript.

NaV and KV channels are miswritten all along the document. They should have a capital V in subscript. Please change them.

As suggested, all "Nav" and "Kv" were changed to "NaV" and "KV" using a capital V in subscript (All are highlighted in yellow).

CFP is no longer called CFP since the toxins have been identified in many other organisms, not only fish. Please check the recent references on the subject. Now, it is called only CP -- ciguatera poisoning.

As suggested, all "ciguatera fish poisoning (CFP)" were changed to "ciguatera poisoning (CP)" (All are highlighted in yellow).

L37- I suggest changing "problem" to "interest". Establishing preventive measures to identify food contaminated with toxins is of major interest. The measures are not "against" ciguatera, but against human intoxication. Please, reword the sentence.

We agreed with the reviewer, and the sentence was changed as suggested (Lines 37–38).

L49-51 - this sentence is miswritten and not clear. Please, rewrite for clarification.

As pointed out, the flow of the sentences was unclear. Therefore, we deleted unnecessary sentences and rearranged the order of the sentences as follows.

<Before change (Lines 49–53)>

CTXs are highly toxic, and based on epidemiological data, the United States Food and Drug Administration (FDA) has proposed that CTX concentrations of 0.01 μg CTX1B eq/kg and 0.1 μg C-CTX-1 eq/kg in fish flesh that are unlikely to cause CFP [30]. Ciguateric fish cannot be judged based on taste or appearance, and CTXs cannot be removed by cooking.

<After change (Lines 50–53)>

The United States Food and Drug Administration (FDA) has proposed that CTX concentrations of 0.01 μg CTX1B equivalent (eq)/kg and 0.1 μg C-CTX-1 eq/kg in fish flesh are unlikely to cause CP [30]. As can be seen from these proposal levels, CTXs are highly toxic. In addition, Ciguateric fish cannot be judged based on taste or appearance.

L116 - it is not "cell death induced by themselves"; it is auto-induced death.

In this experiment, we evaluated cell death induced by KV channel inhibitors. Therefore, "themselves" was changed to "KV channel inhibitors" to avoid ambiguity in "themselves" (Line 117).

L118 - I am not sure of the usefulness of two and three-decimal numbers. It might be easier to read 31.3/3.1

We agreed with the reviewer, and the decimal numbers were changed. However, in order to match the number of digits with 62.5/6.25, it was set as 31.3/3.13 instead of 31.3/3.1 (All are highlighted in yellow).

Figure 1. It is not clear what the x axis is referring to. Please clarify.

Each of the X-axes was clarified by adding "concentration".

Materials and methods: it is preferable to use scientific notation (cells mL-1) than cells/mL

The scientific notation was changed as suggested (Lines 305–306).

In addition to the revisions pointed out by the reviewers, we have tried to reduce the duplication rate as much as possible. They are highlighted in green. On the other hand, many of the duplicates are proper nouns related to experiments and used in previous paper published in Toxicon (2023). Please note that since this paper is an improvement of the method published in Toxicon, a certain amount of duplication is inevitable to ensure that the experiments are accurate.

We hope the revised manuscript is satisfactory for publication in Toxins. Thank you for your kind consideration.

Round 2

Reviewer 3 Report

Comments and Suggestions for Authors

The authord have addressed most of questions raised by the reviewers.